# Study on Score Prediction Model with High Efficiency Based on Deep Learning

**Lihong Yang [1] and Zhiming Bai [2],***

1  Department of Marxism, Hebei Chemical & Pharmaceutical College, Shijiazhuang 050026, China
2  School of Sciences, Hebei University of Science & Technology, Shijiazhuang 050018, China
*  Correspondence: baizhiming@hebust.edu.cn

**Abstract:** In the problem of unified classroom performance prediction, there is a certain lag in the prediction, and there are also problems such as the data sparsity and single feature in the data. In addition, feature engineering is often carried out manually in modeling, which highly depends on the professional knowledge and experience of engineers and affects the accuracy of the prediction to a certain extent. To solve the abovementioned gaps, we proposed an online course score prediction model with a high time efficiency that combines multiple features. The model uses a deep neural network, which can automatically carry out feature engineering and reduce the intervention of artificial feature engineering, thus significantly improving the time efficiency. Secondly, the model uses a factorization machine and two kinds of neural networks to consider the influence of first-order features, second-order features, and higher-order features at the same time, and it fully learns the relationship between the features and scores, which improves the prediction effect of the model compared to using only single feature learning. The performance of the model is evaluated on the learning analysis dataset from Fall 2015 to Spring 2021 and includes 412 courses with 600 students. The experimental results show that the performance of the prediction model based on the feature combination proposed in the present study is better than the previous performance prediction model. More importantly, our model has the best time efficiency of below 0.3 compared to the other models.

**Keywords:** grades prediction; deep learning; data mining; combined feature; factorization machine

## 1. Introduction

In the information age, all walks of life have accumulated a large amount of data, but there is often some useful knowledge and valuable information hidden in the large amount of data. Machine learning and data mining technology can reveal some laws related to data and extract valuable information and data from them, which can be used to solve problems in various fields and provide help for administrators to make more reasonable and effective decisions. At present, machine learning and data mining-related technologies are widely used in the business, finance, medical, and other fields. In conclusion, in the context of education big data, the research on student performance prediction based on deep learning has important scientific significance and application prospects in promoting accurate management, scientific decision making, and improving education quality.

The quality of education is still the focus in the field of education, and improving the quality of education and teaching has always been one of the goals pursued by educators [1]. In the current educational environment, the evaluation of students learning effects is an issue that needs to be solved urgently [2]. In education, student achievement is an important indicator for evaluating the student development potential, development level, and student performance [2]. Student achievement prediction is a process of evaluating and inferring student information in a learning stage by mining and analyzing data based on curriculum settings, student historical grades, and student behavior to discover potentially effective information in teaching data. The predictions include course grades, Grade Point

Average (GPA), course failure risk, and dropout risk [3]. Teachers or administrators can develop or refine teaching strategies, optimize resource allocation, and improve education and educational outcomes based on student abilities. Actions can also be taken based on the expected results. Recently, deep learning technology has made outstanding contributions in all walks of life, especially in the fields of unmanned aerial vehicle swarms [4], real-time systems [5], natural language processing, speech recognition, computer vision, etc. [6,7]. The value obtained by the wide application has also been valued by more and more researchers. Deep learning technology comes from the simulation of information transmission between neurons in the human brain. It extracts basic features from the input, further extracts multi-layer complex features, and learns more feature information. Compared with traditional data mining methods, deep learning algorithms rely on less manual feature extraction and are flexible. Therefore, the application of deep learning technology in student achievement prediction has become one of the important research topics [8].

At present, many scholars have conducted research on the prediction of student academic performance. In the early stage, students' learning data (such as traditional classroom teaching test scores) were mainly collected from the educational administration system, and students' consumption behavior data were collected from the all-in-one card to predict student performance [9]. Some researchers used the learner's interactive records on the platform in the first week and after-school homework performance data, predicting whether a learner will eventually obtain a certificate based on a logistic regression model [10,11]. Brinton and MChiang developed an algorithmic model using factorization machines and K-Nearest Neighbors (KNN) to predict whether a student answered correctly the first time in a massive open online course (MOOC) question [12]. Lorenzo and Gomez-Sanchez utilized logistic regression, stochastic gradient descent, random forest, and support vector machine models to predict whether three engagement metrics (videos, exercises, and assignments) will decline relative to the previous metric at the end of the chapter [13]. Hlosta built a student performance prediction model based on machine learning methods (logistic regression, support vector machine, random forest, naive Bayes, and ensemble learning XGBoost) based on the data generated in the current course to assess whether students are at risk of dropping out of classes [14].

At present, the research on performance prediction has achieved certain results, but there are still some problems and deficiencies. First, in the traditional classroom grade prediction study, the data mainly come from some data generated during the course, such as the course assignment grades, unit test grades, etc. [15]. Therefore, the characteristics required for course grade prediction should be at the end of the course. The prediction results of the final grades are obtained late, the method has a certain lag, and the data also have problems such as sparse data and single features, so it cannot provide effective technical support for the teaching and management work in the early stage of the course [16]. Second, the existing online platform course grade prediction research mainly uses the log data of learners on the learning platform, such as the learners learning time on the online learning platform and the number of clicks on the learning video, and it lacks other relevant course grade information. In addition, the existing research often uses manual feature engineering in the modeling, which is highly dependent on the professional knowledge and experience of the engineers and affects the prediction accuracy of the method to a certain extent [17]. Finally, most of the data used in the existing performance prediction research come from the dataset constructed by the researchers themselves, and the amount of data is generally small. For mainstream research methods such as machine learning algorithms, there are certain requirements for the amount of data. If the amount of data is insufficient, it is difficult to train a better model, which leads to a low prediction accuracy to a certain extent [18].

Educational data mining techniques can provide educational decision makers with data-based models that are essential to support their goals of improving the efficiency and quality of teaching and learning. The main applications of educational data mining



are student performance prediction, the discovery of student bad behavior, knowledge tracking, course recommendation, etc. [19]. Among them, student achievement prediction is a typical problem in educational data mining. Predicting student achievement can help students with learning difficulties in time, and at the same time, it provides teaching suggestions and theoretical support for teaching managers and teachers [20]. Deep learning is a neural network structure based on multiple layers of processing units, which is used in wide fields of practical applications, such as automatic driving [21], path planning [22], natural language processing, and speech recognition. The spread of the deep learning technology allowed college students to create early warning models.

Aiming at the problems of the sparse and lagging prediction data of a traditional classroom in student scores in education data mining, this paper proposed a traditional classroom score prediction model that integrates a self-attention mechanism and depth matrix decomposition, using the course scores learned in previous semesters to predict the course scores to be learned in the next semester or several semesters. First, the self-attention mechanism is added to the model, which can quickly extract the important potential features of the students and courses and make the model more focused on useful information. Secondly, a bilinear pooling layer is built in the model to improve the generalization and learning ability of the model.

As for the problem of online course student performance prediction, the achievement of student performance will be affected by many factors, aiming at the problem that the existing in-depth learning methods of performance prediction do not consider the impact of multiple features on performance prediction at the same time. Therefore, this paper proposed an online course performance prediction model that combines multiple features. Firstly, the model can automatically carry out feature engineering by using a deep neural network, which reduces the intervention of artificial feature engineering. Secondly, the model uses a factorization machine and two kinds of neural networks to consider the influence of first-order features, second-order features, and higher-order features at the same time, fully learning the relationship between the features and grades and improving the prediction effect of the model compared to using only single feature learning.

Our paper is structured as follows. In the Methods and data Section 2, we introduce the method in terms of the deep matrix factorization, dataset, and prediction model of online course grades based on feature combination. The main results of this study are organized in Section 3. Finally, we conclude the main idea in Section 4. Our study highlights that (1) the student achievement prediction model, which combines the self-attention mechanism and depth matrix decomposition, can overcome the problem of lagging prediction results in traditional classroom achievement prediction, and (2) the new feature combination structure model in the present study can overcome the shortcomings of the existing online course score prediction methods.

## 2. Methods and Data

This paper proposed a method to predict student performance, which integrates self-attention mechanism and depth matrix decomposition. Applying the deep matrix decomposition model integrating the self-attention mechanism to predict the course scores that students will learn next semester, similar to the matrix decomposition model.

### 2.1. Deep Matrix Factorization

Recently, however, it was found to be applied to the prediction of student grades. For example, the problem of the prediction of students' curriculum grades in the next semester is analogized to the problem of score prediction or that of the next basket recommendation [23–25]. The method of the recommendation system has brought a new perspective to solve the problem of student grades prediction. In particular, the matrix decomposition method decomposes the student curriculum score matrix into two low-rank matrices to represent the curriculum and student potential knowledge space, and then obtains the scores of the unfinished courses through the dot product of the corresponding vectors in the

two decomposition matrices [26]. The depth matrix decomposition model was successfully applied in the field of recommendation systems. For example, Xue et al. proposed a depth matrix decomposition model to decompose the scoring matrix of users and items and then mapped the decomposed two potential feature vectors to a low-dimensional space through two sets of multi-layer perceptrons, finally realizing the user scoring prediction of movies [27]. Therefore, it can also be used to predict student curriculum achievements.

Assuming $m$ represents number of students and $n$ is the courses number in $T$th semesters, a matrix $G \in R^{m \times n}$ is student–course grades, each item in the $G$ includes a tuple for the student and course. Therefore, each value in $G$ is denoted as $g(s, c)$, which represents the achievement $g(s, c)$ of student $s$ in course $c$, $g(s, c) \in [0, 5]$. In the $T$ semester, the student–course grade (SCG) is expressed as $G^T = \sum_{I=1}^{T} G_i$.

A grade matrix $G$ is constructed by $T$ semester student–course data in the input layer, where $G^{(i)}$ is the grades achieved by students in each course, the $G_j$ is the grades of each student in each course, $(i, j) \in \Omega$. Because the constructed grade matrix is very sparse, the data dimension can be reduced, and the data sparsity can be alleviated through the embedding-layer mapping [28]. For a given score matrix $G_{ij}$, forms of grade matrix are obtained as follows:

$$x_s = Tanh\left(G^i \cdot W_s\right), \ x_c = Tanh\left(G_j \cdot W_c\right) \tag{1}$$

where $x_s$, $x_c \in G^k$ is student latent eigenvectors and course latent eigenvectors, respectively. $W_s \in G^{m \times k}, W_c \in G^{k \times n}$ is the feature projection weight matrix. In the self-attention layer, the student LFV (latent feature vector) and the course LFV are added with different weights. Varied student features and course features have varying importance. The impact of marketing courses on the advanced computer network courses to be studied in the next semester is different. A computer network is the basic subject of advanced computer network, so the relative weight of its influence will be larger, and it will be transformed according to the following formula:

$$x_{sa} = softmax(x_s \cdot W_{sa}) \cdot x_s, \ x_{ca} = softmax(x_c \cdot W_{ca}) \cdot x_c \tag{2}$$

where $W_{sa}$, $W_{ca}$ are self-attention distribution weight parameters. $x_{sa}, x_{sc}$ is the student latent eigenvectors and course latent eigenvectors with weight values, respectively. The multi-layer perceptron layer, it is transformed according to Equation (3), and it aimed to better learn the nonlinear characteristics of the information and perform a two-layer nonlinear mapping between the latent options of the scholars and course.

$$\begin{cases} H_s = sigmoid(x_{sa} \cdot W_s' + b_s) \\ p_s = sigmoid(H_s \cdot W_s'' + b_s') \\ H_c = sigmoid(H_c \cdot W_c'' + b_c) \\ q_c = sigmoid(H_c \cdot W_c'' + b_c') \end{cases} \tag{3}$$

The following transformations are performed in the bilinear pooling layer.

$$\overline{g} = h^T \sigma\left(p_s^T \odot q_c\right) \tag{4}$$

where $h$, $\odot$, $\sigma$ is weight parameter, Hadamard product, activation function, respectively.

The model that predicts the course is built by following:

$$L = min_{W, W_S, W_C, h} \frac{1}{2} \sum_{i,j \in W} \left(G_{ij} - h^T \sigma\left(\left(p_s^{(i)}\right)^T \odot q_{cj}\right)\right)^2 + \lambda(||W_s||_F^2 + ||W_c||_F^2 + ||W||_2) \tag{5}$$

where unknown parameter set, $h^T \sigma\left(\left(p_s^{(i)}\right)^T \odot q_{cj}\right)^2$, is the predicted value of the model output through the nonlinear pooling layer. In addition, learn model parameters with Adams optimization method, using Sigmoid as an activation function.

*2.2. Dataset*

The data used in our work collect all grades from Fall 2015 to Spring 2021 and include 412 courses with 600 students. In the dataset, students course scores range from 0 to 100. Because the predicted target values in the model proposed in present study are discrete, the 0–59 points are converted to 1, the 60–69 points are converted to 2, the 70–79 points are converted to 3, and the 80–89 points are converted to 4. Convert 90–100 points to 5. Train the model on the $(T-1)$th semester dataset and predict the course grades for the $T$th semester. Use the dataset from Autumn 2015 to Autumn 2020 to train the model, and test the model proposed in this chapter on the dataset in Spring 2021 (i.e., $T$ = Spring 2021). The dataset is divided as follows:

Dataset 1: train data are from 2015 Fall to 2020 Fall, test data are 2021 Spring; dataset 2: the train data are from 2015 Fall to 2020 Spring, the test data are 2021 Fall; dataset 3: the train data are from 2015 Fall to 2019 Fall, the test data are 2019 fall.

The method of matrix decomposition can be applied. For example, a performance matrix is decomposed into 2 low-rank matrices containing latent factors of courses and students. $p_s \in G^k$ and $q_c \in G^k$ is students $s$ and courses $c$ latent $k$-dimensional features. Equation (6) can be used to predict the grade of student $s$ in course $c$.

$$\widehat{g_{s,c} = p_s{}^T q_c} \tag{6}$$

The loss function of MF is:

$$L = min_{p,q} \sum_{s,c \in \Omega} \left\| g_{s,c} - p_s q_c{}^{T-1} \right\|^2 + \frac{1}{\lambda}\left( ||p_s||^2 + ||q_c||^2 \right) \tag{7}$$

where $L^2$ is a regular term to prevent overfitting.

*2.3. Prediction Model of Online Course Grades Based on Feature Combination*

Internet, more and more online learning platforms have emerged, and student performance prediction based on online learning behavior data has become a development trend. For the prediction of student grades in online courses, the achievement of student grades will be affected by many factors. It is not as long as the study time is longer or the students who answer the questions better after class can get good grades. Feature combination refers to considering the influence of multiple features on the entire prediction problem at the same time in practical application scenarios.

We proposed a new online course grade prediction method, which is based on different structural combination features, and its main work has three aspects. (1) The study found that considering two or more features at the same time has a certain impact on the prediction results of student grades; (2) the proposed feature combination model can automatically learn the combination of features and consider the effects of first-, second-, and high-order features at the same time; (3) the performance of the model is verified on the open UK Open University learning analysis dataset, which has high accuracy and effectiveness.

Given a set of student features $F$, $F$ consists of student attribute features *student_attr* $= \{s_1, \ldots, s_m\}$, course attributes *course_attr* $= \{c_1, \ldots, c_n\}$, *behavior* $= \{b_1, \ldots, b_k\}$. Among them, $m$, $n$, $k$ are the number of features, respectively. For the student, their final course grade is $y_i$, $y = \{0, 1\}$ is the set of categories divided by the student grade, where 0 means that the student has failed, and 1 means that the student has passed. The goal of student grade prediction is to predict the grade category $y_i$ obtained by students based on student characteristics $F$.

Our study aimed to mine data related to student learning based on deep learning technology, to achieve accurate prediction of student academic performance and to provide timely help and guidance to students who have "risks". Therefore, this study proposes a feature combination-based performance prediction model (factorization deep product neural network, FDPN) based on student attribute characteristics, course attribute characteristics, and student learning behavior characteristics. The model framework is shown

in Figure 1. FDPN consists of 3 layers. Embedding layer: reduce the scale of the initial high-dimensional options and map them to low-dimensional feature vectors. Network layer: this layer consists of factorization machine (FM), deep neural network (DNN), and product neural network (PNN). FM is used to learn the first-order and second-order feature representations of features, and DNN and PNN are, respectively, used to learn higher-order representations between features. Prediction layer: by splicing, the low- and high-order features are combined to obtain the final features with richer information so as to make better student performance predictions.

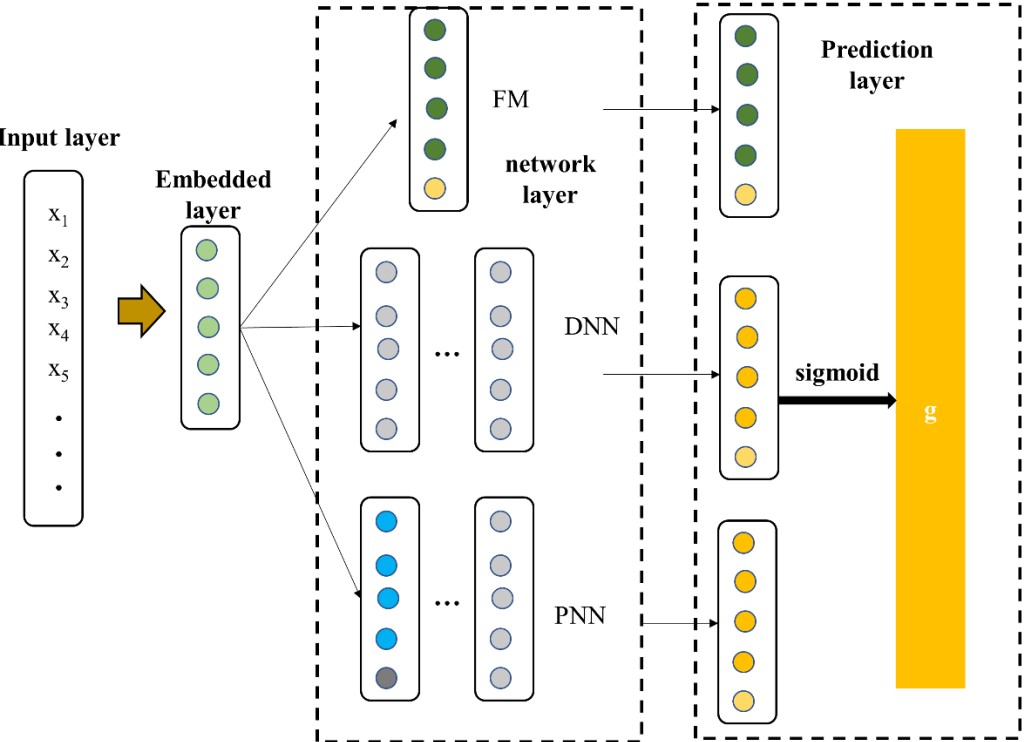

**Figure 1.** Framework of FDPN model. Factorization machine (FM); deep neural network (DNN); product neural network (PNN).

In this paper, the cross entropy loss function is used, and the $L_2$ regularization parameter is used. The loss function of the model is as follows:

$$loss = -\frac{1}{n} \sum_{i=1}^{n} y_i \log g + \lambda ||\theta||^2 \tag{8}$$

Among them, $n$ is the total number of training data, $y_i$ is the grade category of the $i$ data, $g$ is the predicted probability of the grade category of the $i$ data, $\lambda ||\theta||^2$ is $L_2$ regular term, $\theta$ is the set of all parameters of the model.

The evaluation indicators in this paper use accuracy, precision, recall, F1, and AUC (Area Under Curve) to measure the performance of model classification and prediction.

$$\begin{cases} Accuracy = \dfrac{TP + TN}{TP + TN + FP + FN} \\ Precision = \dfrac{TP}{TP + FP} \\ Recall = \dfrac{TP}{TP + FN} \\ F1 = \dfrac{2 \times Precision \times Recall}{Precision + Recall} \end{cases} \tag{9}$$

This paper uses the Open University Learning Analysis Dataset (OULAD), which contains multiple data types, such as the basic information from the Open University in the UK from 2015 to 2021. First, the Open University opens a course, students apply for course registration, and then the course can start to study. The course duration of the Open University is mostly nine months. During the learning process, there will be corresponding learning tasks that the learners must complete.

Three widely used evaluation indicators are used to evaluate the performance of the proposed model.

$$
\begin{aligned}
RMSE &= \sqrt{\frac{1}{\Omega} \sum_{i,j \in \Omega} \left(y_{ij} - \widehat{y_{i,j}}\right)^2} \\
MAE &= \frac{1}{\Omega} \sum_{i,j \in \Omega} \parallel y_{ij} - \widehat{y_{ij}} \parallel \\
MAPE &= \frac{100}{\Omega} \sum_{i=1}^{\Omega} \frac{\parallel y_{ij} - \widehat{y_{ij}} \parallel}{y_i}
\end{aligned}
\tag{10}
$$

where $y_{ij}$ is observations, $\widehat{y_{ij}}$ is average value of observations. *RMSE* is root mean standard error, *MAE* is mean absolute error, *MAPE* is mean absolute percent error. *RM* measures the deviation between the observed value and the true value. *MAE* can better reflect the actual situation of prediction error. *MAPE* can measure the advantages and disadvantages of the model. An *MAPE* of 0% indicates a perfect model, while an *MAPE* greater than 100% indicates a poor quality model.

## 3. Results and Discussion

This experiment trains attention deep matrix factorization (ADFM) models on three datasets. Figure 2 shows the results of the experiments with the ADMF models on three datasets. The ADFM model has the smallest root mean square error (RMSE), mean absolute error (MAE), and mean absolute percent error (MAPE) on dataset 1, and the model prediction effect is the best. In this experiment, the amount of coaching knowledge in the dataset, a pair of, and dataset three is considerably below that in dataset one. Therefore, we will carry out comparative experiments on the dataset.

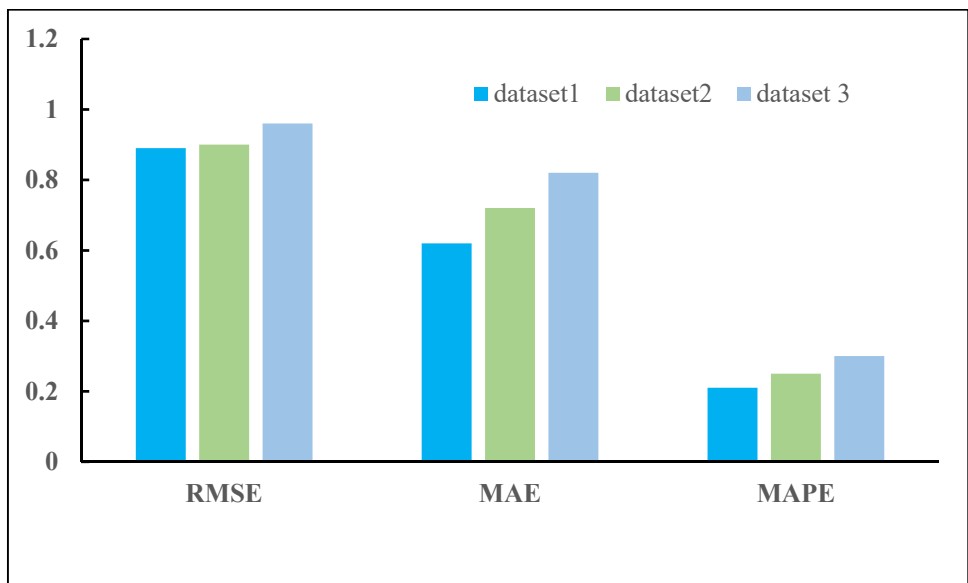

**Figure 2.** DMF prediction effect on different datasets. The description of dataset is introduced in Section 2.2.

From the comparison of the experimental results of the DMF and ADMF in Figure 3, which show that the RMSE of the ADMF model is 84%, the MAE is 66.7%, and the MAPE is

21%, which is smaller than that of the DMF model. Therefore, incorporating a self-attention mechanism into a deep neural network can permit the model to pay much attention to the knowledge which is important for prediction accuracy. During this experiment, the potential feature of the course and students have different weights and different influences on the student performance; therefore, it is clear that the ADMF model contains a sensible impact. The effect of the bilinear pooling layers on the model performance is shown in Table 1.

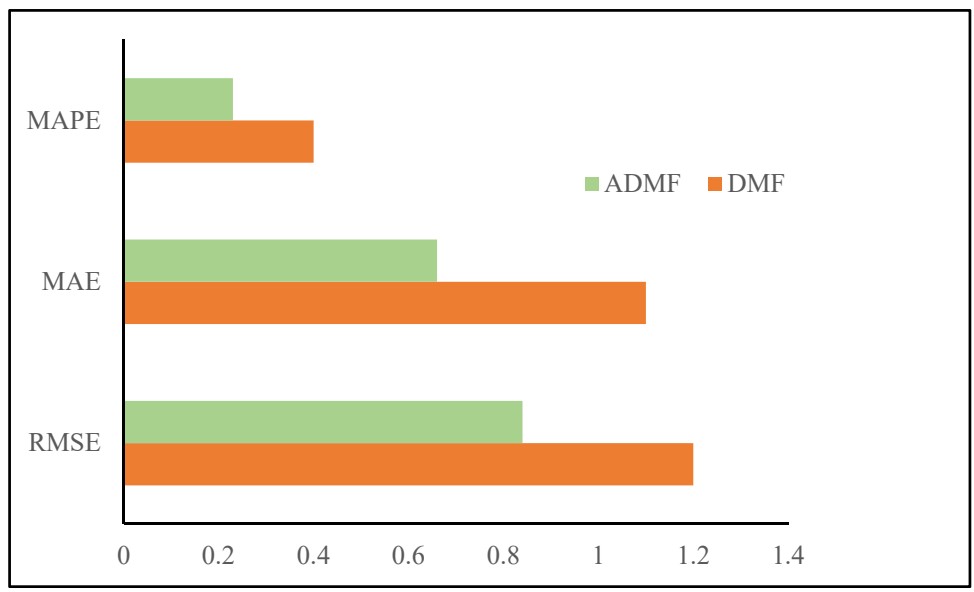

**Figure 3.** Comparison of experimental results between DMF and ADMF.

**Table 1.** Comparison of predicted value output methods.

| Predicted Value Output Method | RMSE | MAE | MAPE |
|---|---|---|---|
| Bilinear pooling | 0.8402 | 0.6523 | 0.2315 |
| Inner product | 0.8478 | 0.6789 | 0.2355 |

We compare two ways to obtain the predicted values, bilinear pooling, and inner product. From the results in Table 1, it is shown that the prediction effect of the bilinear pooling mapping is better than that of the inner product model. The performance of the prediction model proposed in the present study uses the bilinear pooling method to archive the output, which can effectively improve the performance of the model. Regarding the generalization ability, a better performance prediction effect can be obtained.

Several traditional methods of predicting student performance were compared, including the KNN, MF, NCF, and DMF. They were tested on dataset 1, and the final experimental results are shown in Figure 4. We found that the NCF, DMF, and ADMF accomplish higher prediction results than the MF and KNN. The ADMF is the best predictor for students in the next semester. Because the MF is extended to multi-layer neural networks, the ADMF encompasses a stronger ability to spot the input file, which will learn complicated nonlinear structural options hidden in the information, and might conjointly mix easy options into much more complicated options, with flexibility and accuracy. This method decomposes the student course score matrix into row vectors and column vectors, which represent the student scores of each course and all student scores of a specific course, respectively. Then, the row vectors and column vectors are embedded into a low-dimensional space and input into a multi-layer perceptron network based on a self-attention mechanism to further extract information from potential feature vectors, in which the attention mechanism is added, and it can quickly extract the important potential features of the students and courses.

Secondly, a bilinear pooling layer is built in the model to improve the generalization and learning ability of the model.

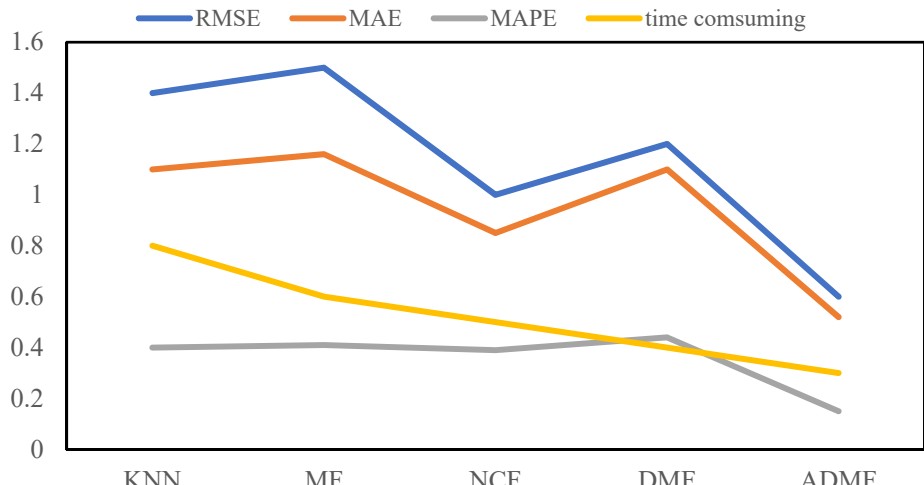

**Figure 4.** Different models predict the effect. Time efficiency is defined by ratio of time consuming for one model to total time for total models.

Our model has two deep neural networks. When the network contains multiple hidden layers, if the number of neurons in the hidden layers is different, many experiments are required. The number of neurons in the layer is set to the same number. The experimental results are shown in Figure 5. This paper carried out six experiments, changing the number of neurons in turn. From the experimental results, when the number of neurons is 256, the recall rate is about 95%, the precision is about 90.7%, the AUC is about 82%, the accuracy rate is about 86.6%, and its model performance is the best. This is mainly because as the number of neurons increases, the model can learn more feature information, but once the variety of neurons increases to a precise number, the model learns the effective information is not increasing, and it will even introduce noise that reduces the prediction performance of the model. It is seen from this that once it involves deep neural network coaching, the quantity of neurons is not the maximum amount as the potential, neither is it as little as the potential. It is necessary to incessantly compare the model coaching and learn to pick out the optimum range of neurons.

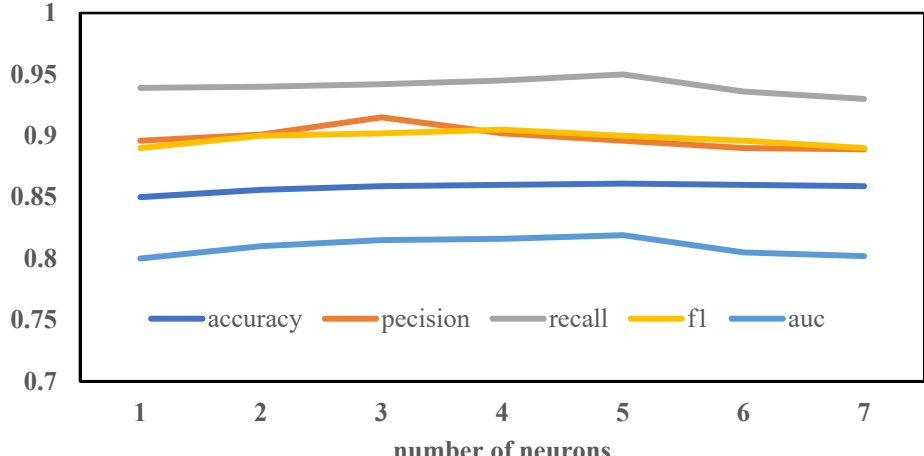

**Figure 5.** Comparison of the number of neural units in the hidden layer.

In our study, we also compared the activation function of the hidden layer neurons with the ReLU activation function and Tanh activation function. The prediction effect of

the model is related to the activation function setting of the hidden neurons. Because this is a binary model, the final prediction output unit of the model uses the sigmoid function, and the other activation functions are set the same. The experimental results are shown in Table 2. It can be seen from the table that when the activation function is the ReLU, the prediction accuracy, recall rate, F1, and AUC of the FDPN model are increased by about 2%, and the prediction effect of the hidden layer neuron activation function using ReLU is better than that of the Tanh function. In the feedforward neural network, the performance of the ReLU activation function will be better than that of the Tanh performance, and the activation performance continues to be set as ReLU within the ensuant experiments of this paper.

**Table 2.** Comparison of activation function.

| Activation Function | Accuracy | Recall | F1 | AUC | Precision |
|---|---|---|---|---|---|
| Tanh | 0.8445 | 0.8610 | 0.9209 | 0.7988 | 0.8610 |
| ReLU | 0.8669 | 0.8680 | 0.9409 | 0.8185 | 0.8680 |

The model proposed in this paper includes two feedforward neural networks, the DNN and PNN. The number of hidden layers in the neural network is different, and the prediction ability of the model is different. To simplify the experiment, this paper sets the same number of hidden layers and neurons in each layer of the two neural networks, that is, the number of hidden layers is increased from one to three. The experimental results are shown in Table 3. It can be seen from the experimental results that when the number of hidden layers is one and the number of hidden layers is two, the model performance is better, and when the number of hidden layers is three, the prediction effect of the model decreases significantly. As the number of layers increases, each evaluation index decreases, mainly because the more layers, the more complex the structure, the greater the amount of calculation, and the problem of the overfitting of the model will occur. Therefore, the number of hidden layers is still set to one in the subsequent experiments of this paper.

**Table 3.** Hidden layer comparison.

| Layers | Accuracy | Precision | Recall | F1 | AUC |
|---|---|---|---|---|---|
| 1 | 0.8668 | 0.8689 | 0.9588 | 0.9012 | 0.8125 |
| 2 | 0.8502 | 0.8632 | 0.9213 | 0.8952 | 0.8050 |
| 3 | 0.7856 | 0.8812 | 0.7852 | 0.8365 | 0.7788 |

The FM learns the first-order and second-order representations of the features, and the DNN and PNN learn different high-order representations of features. In this paper, three structures are combined to learn to predict grades. In the experiments, different feature combination structures are compared to investigate the impact of the structure on the prediction skill of the model. For the results, see Table 4.

**Table 4.** Structures comparison.

| Structure | Accuracy | Precision | Recall | F1 | AUC |
|---|---|---|---|---|---|
| FM | 0.841 | 0.8522 | 0.9279 | 0.8900 | 0.7920 |
| DNN | 0.8439 | 0.8549 | 0.9290 | 0.8910 | 0.7940 |
| DeepFM | 0.8483 | 0.8600 | 0.9397 | 0.8973 | 0.8010 |
| PNN | 0.8586 | 0.8464 | 0.9398 | 0.8004 | 0.8100 |
| DNN + PNN | 0.8641 | 0.8660 | 0.9425 | 0.8937 | 0.8131 |
| FM + PNN | 0.825 | 0.8670 | 0.9440 | 0.9032 | 0.8174 |
| FDPN | 0.8669 | 0.8689 | 0.9499 | 0.9071 | 0.8179 |

The experimental results show that the result of the one-structure FM, DNN, and PNN is worse, the combined feature learning of the DeepFM, DNN + PNN, and FM + PNN of the two structures is higher than that of the one structure, and therefore the combined feature learning result of the three structures of the FDPN model is the best. This is because the FDPN model simultaneously considers the first-order feature representation, the second-order feature representation, and two different high-order feature representations and utilizes more potentially effective information in the grade prediction. To sum up, the FDPN performance prediction model incorporates a smart prediction impact and might improve the performance prediction performance.

In this paper, we use the LR, SVM, FM, and deep learning models such as the DNN, DeepFM, and PNN as the comparison models. Through the comparison experiments on the OULAD dataset, the results are shown in Table 5, which verifies that the FDPN model proposed in this paper has the best prediction. From the experimental results in Table 5, it can be seen that compared with the existing performance prediction methods (LR, SVM, FM, and DNN), the method based on feature combination on the OULAD public education dataset has achieved the best prediction effect. Compared with the best traditional method, the DNN accuracy and AUC are both improved by 2%. In addition, there are also vital enhancements within the three indicators of exactness, recall, and F1. The tactic that supported the feature combination projected during this paper is healthier than the four ancient performance prediction ways within the comparative experiment, mainly as a result of the normal performance prediction methodology which uses every attribute feature directly as a classification feature input model for learning and coaching, and only considers low-level features or high-order features, without considering the different effects of low- and high-order feature combinations on the final grades. For the other two feature combination methods (DeepFM and PNN) in the comparative experiment, the method proposed in this paper extracts more feature information for each attribute feature, including first-order features, second-order features, and two different high-order features. Thereby, the prediction ability of the model is greatly improved, the prediction effect is better, and the validity of the model is finally proved through the experiments.

**Table 5.** Model comparison. Efficiency is defined by 1-(ratio of time consuming for one model to total time for all models.

| Model | Accuracy | Precision | Recall | F1 | Efficiency | AUC |
|---|---|---|---|---|---|---|
| LR [28] | 0.7989 | 0.8339 | 0.8726 | 0.8559 | 0.3 | 0.7552 |
| SVM [29] | 0.8320 | 0.8432 | 0.9225 | 0.8829 | 0.4 | 0.7789 |
| FM [30] | 0.8429 | 0.8465 | 0.9279 | 0.8900 | 0.56 | 0.7939 |
| DNN [31] | 0.8439 | 0.8552 | 0.9290 | 0.8910 | 0.61 | 0.8008 |
| DeepFM [32] | 0.8480 | 0.8623 | 0.9289 | 0.8940 | 0.55 | 0.8002 |
| PNN [33] | 0.8546 | 0.8656 | 0.9389 | 0.9002 | 0.75 | 0.8089 |
| FDPN | 0.8658 | 0.8679 | 0.9498 | 0.9069 | 0.86 | 0.8180 |

## 4. Conclusions

(1) This paper proposed a student performance prediction model that integrates a self-attention mechanism and depth matrix decomposition. The experimental results show that the method proposed in this chapter is superior to the benchmark comparison method. The RMSE of the ADMF model is 84%, the MAE is 66.7%, and the MAPE is 21%, which is smaller than that of the DMF model.

(2) A new feature combination structure model is proposed to overcome the shortcomings of the existing online course score prediction methods. The experimental results show that the model proposed in this study has a good performance prediction ability. When the number of neurons is 256, the recall rate is about 95%, the precision is about 90.7%, the AUC is about 82%, and the accuracy rate is about 86.6%.

However, it does not consider the impact of the time series of students learning courses, student comments on online platforms, and the other contents on student scores. (i) Because

of the traditional classroom performance prediction problem, this paper only uses the final scores of multiple courses to collect more student-related data, such as the number of times students borrow books every week, relevant information about teachers, etc., considering the impact of multiple characteristics on the student curriculum performance, and further considering the relationship between courses. (ii) For the online platform course score prediction problem, we can consider more in-depth learning algorithm models, consider the different effects of different features, and use the attention mechanism to learn their different weights to further improve the prediction performance of the model. (iii) For the above two aspects of the performance prediction, the time series of course learning has not been considered. In future work, we can consider the influence of this factor to further explore.

**Author Contributions:** Conceptualization, methodology, validation, formal analysis, investigation, resources and writing, L.Y.; data curation, writing—original draft preparation, review and editing, Z.B. All authors have read and agreed to the published version of the manuscript.

**Funding:** This research received no external funding.

**Data Availability Statement:** The data used to support the findings of this study are available from the corresponding author upon request.

**Conflicts of Interest:** We declare that we have no financial and personal relationships with other people or organizations that can inappropriately influence our work. There is no professional or other personal interest of any nature or kind in any product, service, and/or company that could be construed as influencing the position presented in, or the review of, the manuscript entitled.

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
