# Peer review of "Study on Score Prediction Model with High Efficiency Based on Deep Learning"

_electronics, doi:10.3390/electronics11233995_

Round 1
Reviewer 1 Report
The authors present the article entitled “Study on college student grades prediction model with high efficiency based on deep learning”
This paper proposes a traditional classroom score prediction model that integrates the self-attention mechanism and depth matrix decomposition, using the course scores learned in previous semesters to predict the course scores learned in the next semester or several semesters.
The article presents the following concerns:
-
It is recommendable to describe the structure of the text at the end of the introduction.
-
Add a little introduction between sections 2 and 2.1
-
Add hyperlinks to tables, figures, and references
-
Unifies the font size throughout the document
-
Apostrophes must be avoided, for example: students'
-
Can you explain which is the relevance of the topic vs the aim of the journal?
-
Line 31 can be justified with the following references about the quality education: Constructivism-based methodology for teaching artificial intelligence topics focused on sustainable development; An approach to STFT and CWT learning through music hands-on labs.
-
Line 54 can be justified with up-to-date references regarding student and teacher's academic performance.
The following misspelling should be checked:
-
title: The phrase “college student grades prediction model” appears to be a confusing noun string. Consider rewriting the sentence for clarity.
-
line 17: “improve significantly time efficiency…” should be rewritten by: “significantly improving time efficiency…”
-
line 82: The use of “etc.” informal writing is generally frowned upon.
-
line 100: The phrase “on the basis of” may be wordy. Consider changing by “based on”
-
line 103: The phrase “provides the chance for” may be wordy. Consider changing by “allows”
-
line 306: The phrase “a lot of” may be considered for formal writing. Consider changing by “much”
-
line 326: “Will increase” seems to be in the wrong tense. Changing by “increases”
-
line 329: Contractions such as “isn’t” may be too informal for this writing style. Consider replacing it with an uncontracted form.
-
line 436: The phrase “In view of” may be wordy. Consider changing the wording by “Given” or “because of”
Author Response
Reviewer#1 The authors present the article entitled “Study on college student grades prediction model with high efficiency based on deep learning” This paper proposes a traditional classroom score prediction model that integrates the self-attention mechanism and depth matrix decomposition, using the course scores learned in previous semesters to predict the course scores learned in the next semester or several semesters. The article presents the following concerns: • It is recommendable to describe the structure of the text at the end of the introduction. We have added the sentence referring to structure of paper. Our paper is structed by follows. In the Methods and data Sec., we introduced the method in terms of deep matrix factorization, data set and prediction model of online course grades based on feature combination. The main results of study are organized in Sec.3. Finally, we concluded the main idea in Sec.4. • Add a little introduction between sections 2 and 2.1 We have added the related sentence. This paper proposes a method to predict students' performance, which integrates self attention mechanism and depth matrix decomposition. Applying the deep matrix decomposition model integrating the self attention mechanism to predict the course scores that students will learn next semester, similar to the matrix decomposition model • Add hyperlinks to tables, figures, and references Thank you for your careful comment, it is more convenient for reading to hyperlink to tables/figures/references, we have complete it in the revised version. • Unifies the font size throughout the document Thanks again for your careful comments, we write our paper according to mode of MDPI, the format of text including size, font is unified throughout the paper. • Apostrophes must be avoided, for example: students' We have revised them in the revised paper. Can you explain which is the relevance of the topic vs the aim of the journal? Thank you for your question. Aiming at the problem that the existing deep learning methods of score prediction do not consider the influence of multiple features on score prediction at the same time, this paper proposes an online course score prediction model that combines multiple features. First of all, the model can automatically carry out feature engineering by using deep neural network, which reduces the intervention of artificial feature engineering. Secondly, the model uses factorization machine and two kinds of neural networks to consider the influence of first-order features, second-order features and higher-order features at the same time, fully learning the relationship between features and grades, and improving the prediction effect of the model compared to using only a single feature learning. The topic of our study is deep learning, big data analysis and application in school students, which is related to the topic for special issue of Journal in terms of deep learning and big data analysis and applications in other fields. Line 31 can be justified with the following references about the quality education: Constructivism-based methodology for teaching artificial intelligence topics focused on sustainable development; An approach to STFT and CWT learning through music hands-on labs. we are very agree with your comments about this two references. The first reference “Constructivism-based methodology for teaching artificial intelligence topics focused on sustainable development” proposed the creation of a course based on a series of practical sessions, where the students have to develop their practical knowledge about artificial intelligence techniques, specifically multilayer perceptron, which is oriented to evaluate skills in the broad education necessary to understand the impact of engineering solutions in a global, economic, environmental, and societal context. The proposal helps the students to turn theoretical concepts into more tangible objects where they can build their knowledge by programming their implementations in software. The second reference, “An approach to STFT and CWT learning through music hands-on labs”, presented a series of lectures and activities for Digital Signal Processing (DSP) teaching, based upon music and their principal elements such as melody, pitch, timbre, beat, and metric, to explain the time-frequency analysis and its repercussions in other areas of engineering. In order to justify the results from our results, we citied these two references in our paper. Line 54 can be justified with up-to-date references regarding student and teacher's academic performance. We try our best to find up-to-date reference to justify our comments in terms of students performance. The findings of the Ucar’s study offer insights into ARCSV model‐based research by examining the effects of the model as a valid and reliable framework for online learning environments. The implications and directions for future research are then discussed. We cited it in our revised paper to support our comments. The following misspelling should be checked: title: The phrase “college student grades prediction model” appears to be a confusing noun string. Consider rewriting the sentence for clarity. We have improved the title. Study on score prediction model with high efficiency based on deep learning. line 17: “improve significantly time efficiency…” should be rewritten by: “significantly improving time efficiency…” we have rewritten it. …thus significantly improve time efficiency… line 82: The use of “etc.” informal writing is generally frowned upon. We have removed it. line 100: The phrase “on the basis of” may be wordy. Consider changing by “based on” we have revised it. Deep learning is a neural network structure based on multiple layers of processing units, line 103: The phrase “provides the chance for” may be wordy. Consider changing by “allows” we have improved it. The spread of deep learning technology allowed college kids to create early warning models. line 306: The phrase “a lot of” may be considered for formal writing. Consider changing by “much” we have rewritten it into much. line 326: “Will increase” seems to be in the wrong tense. Changing by “increases” we have improved it. …the model can learn more feature information, but once variety of neurons increase to a precise number… line 329: Contractions such as “isn’t” may be too informal for this writing style. Consider replacing it with an uncontracted form. \ we have improved it. It is seen from this that once it involves deep neural network coaching, the quantity of neurons is not the maximum amount as potential, neither is it as little as potential. It is necessary to incessantly compare the model coaching and learning to pick out the optimum range of neurons. line 436: The phrase “In view of” may be wordy. Consider changing the wording by “Given” or “because of” thank you for your careful suggestions, we have improved it. Because of the traditional classroom performance prediction problem, this paper only us-es the final scores of multiple courses to collect more student related data.

Reviewer 2 Report
Extensive editing of the English language and style must be performed.
Lots of sentences have repeated words (3 or more times).
Some sentences are without sense, such as in lines 272-278.
Some acronyms are not defined the first time they appear (MOOC, ADMF). ADMF sometimes appears as ADFM.
The experimental part of the work is not sufficiently explained. How datasets 1 to 3 were formed? We need more detail on data. Which dataset is present in each figure or table? Dataset 1 (the best), a union of the 3 datasets? You need to describe the conditions of the experiments much better.
Conclusions are also very vague and should be clearer about the strengths and weaknesses of the results.
Author Response
Reviewer#2
Extensive editing of the English language and style must be performed.
Lots of sentences have repeated words (3 or more times).
Some sentences are without sense, such as in lines 272-278.
Some acronyms are not defined the first time they appear (MOOC, ADMF). ADMF sometimes appears as ADFM.
Thank you for your efforts for English grammar checking, we have checked our paper’s grammar entirely, and improved them in the revised version, if there are still grammar mistakes, pleas let us known it.
In recent years, with the development of the Internet and the continuous improvement of online learning platforms, student learning-related data in massive open online course (MOOC) has obtained more attention [10].
This experiment trains attention deep matrix factorization (ADFM) models on three datasets.
Line 272-278: This experiment trains attention deep matrix factorization (ADFM) models on three datasets. Fig.2 shows the results of experiments with ADMF models on three datasets. The ADMF model has the smallest root mean square error (RMSE), mean absolute error (MAE), and mean absolute percent error (MAPE) on dataset 1, and the model prediction effect is the best. In this experiment, the amount of coaching knowledge in dataset, a pair of and dataset three is considerably below that in dataset one. The analysis of the on top of results show that, on the 3 datasets, the much the quantity of information, the higher the prediction result of the ADMF model. Therefore, we will carry out comparative experiments on dataset.
The experimental part of the work is not sufficiently explained. How -datasets 1 to 3 were formed? We need more detail on data. Which dataset is present in each figure or table? Dataset 1 (the best), a union of the 3 datasets? You need to describe the conditions of the experiments much better.
Thank you, we are sorry to fail to present our dataset clearly to you, now, we have rewritten the Dataset Sec. in the revised paper. firstly, we trained attention deep matrix factorization (ADFM) models on three datasets (dataset 1-3), results in Fig.2 show that the performance of model in present study is the best, thus dataset 1 is analyzed in the comparison of experimental between DMF and ADMF. Secondly, The Open University Learning Analysis Dataset is used to predict the performance of the Factorization Deep Product Neural network (FDPN). To sum up, (1) a series of comparative experiments have been done on the self-built dataset to prove the effectiveness of the proposed model (ADMF); (2) The accuracy of the proposed feature combination model is verified on the open UK Open University learning analysis data set. We have explained them in the each result section.
The data used in our work collects all grades from fall 2015 to spring 2021 and includes 412 courses with 600 students. In the dataset, students course scores range from 0 to 100. Since the predicted target values in the model proposed in present study are dis-crete, the 0-59 points are converted to 1, the 60-69 points are converted to 2, the 70-79 points are converted to 3, and the 80-89 points are converted to 4. Convert 90-100 points to 5. Train the model on the (? – 1)th semester dataset and predict the course grades for the ?th semester. Use the data set from autumn 2015 to autumn 2020 to train the model, and test the model proposed in this chapter on the data set in spring 2021 (i.e. ?=spring 2021). The data set is divided as follows:
Dataset 1: train data is from 2015 fall to 2020 fall, test data is 2021 spring; dataset 2: the train data is from 2015 fall to 2020 spring, the test data is 2021 fall; dataset 3 :the train data is from 2015 fall to 2019 fall, the test data is 2019 fall.
Conclusions are also very vague and should be clearer about the strengths and weaknesses of the results.
Thanks again, we have improved it in the revised paper.
(1) This paper proposed a student performance prediction model that integrates self-attention mechanism and depth matrix decomposition. The experimental results show that the method proposed in this chapter is superior to the benchmark comparison method. RMSE of the ADMF model is 84%, MAE is 66.7%, and MAPE is 21%, that is smaller than that of DMF model.
(2) A new feature combination structure model is proposed to overcome the short-comings of existing online course score prediction methods. The experimental results show that the model proposed in this study has good performance prediction ability. When the number of neurons is 256, the recall rate is about 95%, about 90.7%, the AUC is about 82%, the accuracy rate is about 86.6 %, the accuracy rate is 86.8%.
However, it does not consider the impact of time series of students learning courses, student comments on online platforms and other contents on student scores. (i) Because of the traditional classroom performance prediction problem, this paper only uses the final scores of multiple courses to collect more student related data, such as the number of times students borrow books every week, relevant information about teachers, etc., considering the impact of multiple characteristics on student curriculum performance, and further considering the relationship between courses. (ii) For the online platform course score prediction problem, we can consider more in-depth learning algorithm models, con-sider the different effects of different features, and use the attention mechanism to learn their different weights to further improve the prediction performance of the model. (iii) For the above two aspects of performance prediction, the time series of course learning has not been considered. In future work, we can consider the influence of this factor to further explore.

Reviewer 3 Report
The concept of the paper is very good. The authors have presented very important and real-time topics in their research work.
1. The introduction part need to be modified with incorporation of the background study.
2. Mention the main highlights of the paper at the end of the introduction part in bulleted form.
3. Mention the novelty of the work.
4. Clarify about the advantages of presented approach compared with the application of machine learning method.
5. Check the equation format once again.
6. Is Fig. 1 your own contribution? Otherwise mention the references.
7. The efficiency of FDPN in Table 5 shows very high as compared to the other method. Clearly mention the reasons.
8. Modify the conclusion part by including some numerical results.
Author Response
Reviewer#3
The concept of the paper is very good. The authors have presented very important and real-time topics in their research work.
- The introduction part need to be modified with incorporation of the background study.
we have added a sentence referring to background as follows.
In the information age, all walks of life have accumulated a large amount of data, but there are often some useful knowledge and valuable information hidden in the large amount of data. Machine learning and data mining technology can reveal some laws re-lated to data, and extract valuable information and data from them, which can be used to solve problems in various fields and provide help for administrators to make more rea-sonable and effective decisions. At present, machine learning and data mining related technologies are widely used in business, finance, medical and other fields. In conclusion, in the context of education big data, the research on student performance prediction based on deep learning has important scientific significance and application prospects in promoting accurate management, scientific decision-making and improving education quality.
- Mention the main highlights of the paper at the end of the introduction part in bulleted form.
Thank you for your suggestions. We have added what you recommended.
Our study highlighted that (1) the student achievement prediction model, which com-bines the self-attention mechanism and depth matrix decomposition, can overcome the problem of lagging prediction results in traditional classroom achievement prediction; (2) the new feature combination structure model can overcome the shortcomings of existing online course score prediction methods.
- Mention the novelty of the work.
We have mentioned it in our last paragraph of the Introduction section.
Aiming at the problems of sparse and lagging prediction data of traditional class-room in student scores in education data mining, this paper proposed a traditional class-room score prediction model that integrates self-attention mechanism and depth matrix decomposition, using the course scores learned in previous semesters to predict the course scores to be learned in the next semester or several semesters. First, the self-attention mechanism is added to the model, which can quickly extract the important potential features of students and courses, and make the model more focused on useful information. Secondly, a bilinear pooling layer is built in the model to improve the generalization and learning ability of the model.
As for the problem of online course student performance prediction, the achievement of student performance will be affected by many factors: aiming at the problem that the existing in-depth learning methods of performance prediction do not consider the im-pact of multiple features on performance prediction at the same time. Therefore, this paper proposed an online course performance prediction model that combines multiple features. Firstly, the model can automatically carry out feature engineering by using deep neural network, which reduces the intervention of artificial feature engineering. Secondly, the model uses factorization machine and two kinds of neural networks to consider the in-fluence of first-order features, second-order features, and higher-order features at the same time, fully learning the relationship between features and grades, and improving the pre-diction effect of the model compared to using only a single feature learning.
- Clarify about the advantages of presented approach compared with the application of machine learning method.
Thank you! In fact, we have added it in our Introduction section.
Our paper is structed by follows. In the Methods and data Sec., we introduced the method in terms of deep matrix factorization, data set and prediction model of online course grades based on feature combination. The main results of study are organized in Sec.3. Finally, we concluded the main idea in Sec.4. Our study highlighted that (1) the student achievement prediction model, which combines the self-attention mechanism and depth matrix decomposition, can overcome the problem of lagging prediction results in traditional classroom achievement prediction; (2) the new feature combination structure model in present study can overcome the shortcomings of existing online course score prediction methods.
- Check the equation format once again.
Thank you for your suggestions. We have improved them in the revised paper.
- Is Fig. 1 your own contribution? Otherwise mention the references.
Fig. 1 is come from our group, which clarified framework of FDPN model in a plain image.
- The efficiency of FDPN in Table 5 shows very high as compared to the other method. Clearly mention the reasons.
We have discussed the reason that why FDPN model have the best performance in line 398-409.
The tactic supported the feature combination projected during this paper is healthier than the four ancient performance prediction ways within the comparative experiment, mainly as a result of the normal performance prediction methodology uses every attribute feature directly as a classification feature input model for learning and coaching, and only con-siders low-level features or high-order features, without considering the different effects of low- and high-order feature combinations on final grades. For the other two feature com-bination methods (DeepFM, PNN) in the comparative experiment, the method proposed in this paper extracts more feature information for each attribute feature, including first-order features, second-order features, and two different high-order features, thereby the prediction ability of the model has been greatly improved, and the prediction effect is better, and the validity of the model is finally proved through experiments.
- Modify the conclusion part by including some numerical results.
We have improved it.
(1) This paper proposed a student performance prediction model that integrates self-attention mechanism and depth matrix decomposition. The experimental results show that the method proposed in this chapter is superior to the benchmark comparison method. RMSE of the ADMF model is 84%, MAE is 66.7%, and MAPE is 21%, that is smaller than that of DMF model.
(2) A new feature combination structure model is proposed to overcome the short-comings of existing online course score prediction methods. The experimental results show that the model proposed in this study has good performance prediction ability. When the number of neurons is 256, the recall rate is about 95%, about 90.7%, the AUC is about 82%, the accuracy rate is about 86.6 %, the accuracy rate is 86.8%.

Round 2
Reviewer 1 Report
The manuscript has been improved.
Author Response
Thank you for affirming my manuscript, and thank you for your valuable guidance. At the same time, I carefully modified the language of the article again.
Reviewer 2 Report
There are still some errors, even in new text:
- line 133: Our paper is structed by follows. Should be: Our paper is structured as follows.
- line 287: should be ADMF instead of ADFM.
- lines 291-292: sentence does not make sense.
There is a reference broken link in line 64 (Error! Reference source not found.)
Methodology, results, and conclusions were improved in this latest version. However, a complete revision of the language and grammar is still necessary.
Author Response
We are so grateful the reviewer’s comments for our manuscript, the comments are related to our English grammar. We improved our paper in the second version which are marked as red in the revised version. Details point-by point is given as follows. Where reviewer’s comments are marked as blue, the answer is marked black.
Reviewer#1
There are still some errors, even in new text:
- line 133: Our paper is structed by follows. Should be: Our paper is structured as follows.
We have revised it according to your suggestion
- line 287: should be ADMF instead of ADFM.
We have changed it.
- lines 291-292: sentence does not make sense.
We deleted it.
There is a reference broken link in line 64 (Error! Reference source not found.)
We have changed the reference [9].
Methodology, results, and conclusions were improved in this latest version. However, a complete revision of the language and grammar is still necessary.
Thank you for your careful comments. We have checked our English grammar thoroughly, forgive our poor English skill as non-native researcher. If you still find mistakes, please let us know that.